# Hydroxylation of Progesterone and Its Derivatives by the Entomopathogenic Strain *Isaria farinosa* KCh KW1.1

**DOI:** 10.3390/ijms23137015

**Published:** 2022-06-24

**Authors:** Ewa Kozłowska, Jordan Sycz, Tomasz Janeczko

**Affiliations:** Department of Food Chemistry and Biocatalysis, Wrocław University of Environmental and Life Sciences, Norwida 25, 50-375 Wrocław, Poland; jordan.sycz@upwr.edu.pl

**Keywords:** biotransformation, progesterone derivatives, entomopathogenic fungi, *Isaria farinosa*

## Abstract

Progesterone biotransformation is worth studying because of the high industrial value of its derivatives. This study investigated the catalytic ability of the entomopathogenic filamentous fungus strain *Isaria farinosa* KCh KW1.1 to transform progesterone derivatives: 11α-hydroxyprogesterone, 17α-hydroxyprogesterone, 16α,17α-epoxyprogesterone and pregnenolone. In the culture of *Isaria farinosa* KCh KW1.1, 11α-hydroxyprogesterone was effectively transformed into only one product: 6β,11α-dihydroxyprogesterone. Transformation of 17α-hydroxyprogesterone gave three hydroxy derivatives: 6β,17α-dihydroxyprogesterone, 12β,17α-dihydroxyprogesterone and 6β,12β,17α-trihydroxyprogesterone. Two products: 6β-hydroxy-16α,17α-epoxyprogesterone and 6β,11α-dihydroxy-16α,17α-epoxyprogesterone, were obtained from the 16α,17α-epoxyprogesterone transformation. We isolated two compounds from the biotransformation medium with pregnenolone: 11α-hydroxy-7-oxopregnenolone and 5α,6α-epoxy-3β,11α-dihydroxypregnan-7,20-dione. In this study, we observed only mono- and dihydroxy derivatives of the tested substrates, and the number of obtained products for each biotransformation did not exceed three.

## 1. Introduction

Progesterone biotransformation is well studied because of the high industrial value of its derivatives, especially 11α-hydroxyprogesterone. The discovery of cortisone’s anti-inflammatory properties and efficient 11α-hydroxylation catalysed by a *Rhizopus* species strain has intensified interest in progesterone biotransformations. In order to maximise the process of obtaining hydroxy steroid derivatives, many procedures are carried out, e.g., optimisation of biotransformation, increasing substrate solubility, searching for cheaper raw material from which the initial steroid compound may be obtained, genetic modification of previously used microorganisms, optimisation of the purification process or the most recent strategy—protein engineering [1,2,3,4,5,6,7,8]. The search for new microorganisms capable of efficient steroid transformation also fits this trend.

Entomopathogenic fungi are filamentous fungi that infect insects and are classified into different taxa. Due to their unique environment, they produce a range of enzymes. The best known steroid catalysts in this group are the genera *Beauveria* and *Metarhizium*. Our previous research has provided information about new steroid catalysts that belong to this group, namely strains of *Isaria farinose*, *Isaria fumosorosea* and *Beauveria caledonica* [9,10,11,12]. Among the many tested entomopathogenic filamentous fungi, the *Isaria farinosa* KCh KW1.1 strain was characterised by efficient transformation of progesterone (**1**) into 6β,11α-dihydroxyprogesterone (**2**), which was obtained with high conversion as the main product [9].

*Isaria farinose*, formerly known as *Paecilomyces farinosus*, is an important entomopathogenic fungus with a worldwide distribution and multiple host insects [13,14,15]. In addition to the ability of effective hydroxylation of progesterone, the transformation of dehydroepiandrosterone (DHEA) in the culture of twelve *Isaria farinosa* strains led to the formation of 7α- and 7β-hydroxy-DHEA [9]. We also observed the capacity of the *Isaria farinosa* KCh KW1.1 strain for effective epoxidation of 17β-hydroxyandrost-1,4,6-triene-3-one [9]. During the transformation of 3β-acetyloxy-5α-chloro-6,19-oxidoandrostan-17-one in the culture of the same strain, we obtained three 11α-hydroxy derivatives. They are the result of consecutive hydrolysis of the acyl group at C-3, 11α-hydroxylation (3β,11α-dihydroxy-5α-chloro-6,19-oxidoandrostan-17-one), followed by hydroxylation of the C-19 carbon atom (3β,11α,19-trihydroxy-5α-chloro-6,19-oxidoandrostan-17-one) and its further oxidation, leading to the formation of lactone (3β,11α-dihydroxy-5α-chloro-6,19-oxidoandrostan-17,19-dione) [16]. Strains from this species have a unique capacity for 4-*O*-methylglycosylation of flavonoids. This reaction is most often described for flavonoid compounds containing a free hydroxyl group in their structure [17]. The ability of *Isaria farinosa* strains to hydroxylate and 4-*O*-methylglycosylate flavonoid compounds was also described [18,19,20]. In our previous work, we also observed that they perform demethylation and then 4-*O*-methylglycosylation of methoxyflavones [19,21].

This study investigated the catalytic ability of the entomopathogenic filamentous fungus strain *Isaria farinosa* KCh KW1.1 to transform progesterone derivatives. We also provided computational analysis of compounds using the free web tool SwissADME, which evaluates pharmacokinetics and drug likeness of small compounds [22].

## 2. Results

In our previous study regarding biotransformations by *Isaria farinosa* strains, progesterone was one of the model steroids [9]. In the culture of *Isaria farinosa* KCh KW1.1, progesterone was effectively transformed into only two products: 6β,11α-dihydroxyprogesterone (**2**) and 6β-hydroxy-11-oxoprogesterone (**7**) (Figure 1). The rapid and highly efficient process and one leading product (**2**; 75% yield after 24 h, established by GC) prompted us to conduct further research on this strain’s biotransformation of progesterone derivatives.

After a 3-day incubation of 100 mg of progesterone (**1**) in *Isaria farinosa* KCh KW1.1 culture, 32.9 mg of 6β,11α-dihydroxyprogesterone (**2**) and 4.1 mg of 6β-hydroxy-11-oxo-progesterone (**7**) were isolated.

Progesterone was transformed into 6β,11α-dihydroxyprogesterone (**2**), but under the conditions of the previous experiment, no intermediate product (6β- or 11α-hydroxyprogesterone) was observed [9]. To verify the possible sequence of hydroxylation, 11α-hydroxyprogesterone (**3**) was used as a substrate. As we assumed, it was also hydroxylated at the 6β position (Figure 2). The product was assigned based on comparing its NMR data and R_f_ and R_t_ on the TLC and GC with the previous study’s compound [9]. No oxidation product of the 11α-hydroxyl group was isolated. The other possible intermediate of progesterone transformation, 6β-hydroxyprogesterone, is not commercially available, so the possibility of hydroxylation of a 6β-hydroxy derivative cannot be excluded.

For a better understanding of the catalytic abilities of *Isaria farinosa* KCh KW1.1, another hydroxy derivative, 17α-hydroxyprogesterone (**4**), was used as a substrate. In this case, we isolated three hydroxy derivatives: 6β,17α-dihydroxyprogesterone (**8**), 12β,17α-dihydroxyprogesterone (**9**) and 6β,12β,17α-trihydroxyprogesterone (**10**) (Figure 3). All the compounds were determined based on their NMR data. In the ^1^H spectrum of **8**, in contrast to the 17α-hydroxyprogesterone (**4**), a signal at δ_H_ = 4.44 ppm appears, which shows the appearance of a new hydroxyl group. Chemical shifts of characteristic methyl group signals (C-18, C-19 and C-21) and 4-H indicate that the isolated compound is 6β,17α-dihydroxyprogesterone (**8**). In addition, the structure of this compound confirms agreement with the literature data [23,24,25].

In the ^13^C spectrum of 12β,17α-dihydroxyprogesterone (**9**), in contrast to the substrate spectrum, there is also a signal from a new hydroxyl group (δ_C_ = 70.08 ppm), which is bound to a proton at δ 3.94 ppm (dd), and the COSY spectrum shows the coupling between this resonance and 11-Hα and 11-Hβ. Spectral data obtained for this compound are in high agreement with previously published data [24]. In the ^13^C spectrum of 6β,12β,17α-trihydroxyprogesterone (**10**), three hydroxyl groups appear at δ_C_ = 73.14 and 89.22 bound to protons at δ_H_ = 4.37 (triplet) and 3.98 (dd), respectively. Chemical shifts of the characteristic signals of the methyl groups (C-18, C-19 and C-21) and 4-H and comparison of the results of NMR analyses performed for this compound with the literature data [24] confirm that compound **10** is a trihydroxy derivative of progesterone—6β,12β,17α-trihydroxyprogesterone (**10**).

The 17α-hydroxyprogesterone (**4**) was also hydroxylated at the 6β position, which seems to be characteristic of progesterone transformation by this strain. The polar hydroxyl group at the C-17α in this substrate must interact with the enzyme’s active site because the other hydroxylation did not occur at the 11α position but at the 12β.

Additionally, another substrate with oxygen bonded to C-17, i.e., 16α,17α-epoxyprogesterone (**5**), was used (Figure 4). Similarly to 17α-hydroxyprogesterone (**4**), it has an oxygen group from the α-face. However, the results of transformation **5** are slightly different. Two products, 6β-hydroxy-16α,17α-epoxyprogesterone (**11**) and 6β,11α-dihydroxy-16α,17α-epoxyprogesterone (**12**), were obtained. In the ^1^H spectrum of **11**, new signals appear at δ_H_ = 4.35 (triplet) from a new hydroxyl group (spectral data consistent with literature data [26]). The ^13^C spectrum of **12** reveals two new hydroxyl groups at 73.04 and 68.55 ppm (Table 1) and the unchanged 16α,17α-epoxy moiety. The structure of this product is confirmed by both the multiplet position of proton-derived signals on the ^1^H NMR spectrum, the positions of carbon-derived signals on the ^13^C NMR spectrum and the correlation spectra (Appendix A).

Hydroxylation at the 6β position occurred, indicating that such steric bulk as a hydroxyl or epoxy group at C-17 does not influence this carbon atom’s hydroxylation. Surprisingly, the other hydroxylation did not occur at the 12β position but at 11α.

After a 10-day biotransformation of 16α,17α-epoxyprogesterone in the culture of *Isaria frinose* KCh KW1.1, 4 mg of 6β-hydroxy-16α,17α-epoxyprogesterone and 25.9 mg of 6β,11α-dihydroxy-16α,17α-epoxyprogesterone were isolated.

In contrast to 3-oxo-4-ene substrates (progesterone derivatives), we also tested pregnenolone (**6**), a C-21 steroid with a 3β-hydroxy-5-ene moiety. We isolated two compounds from the biotransformation medium with the *Isaria farinosa* KCh KW1.1 strain, 11α-hydroxy-7-oxopregnenolone (**13**) and 5α,6α-epoxy-3β,11α-dihydroxypregnan-7,20-dione (**14**) (Figure 5). Both products have a hydroxyl group at the 11α position, as progesterone derivatives have (excluding 17α-hydroxyprogesterone) a 7-oxo moiety. In our previous study, dehydroepiandrosterone, an analogous C-19, 3β-hydroxy-5-ene steroid, was hydroxylated only at the 7α and 7β positions [9]. With a very high probability, we can assume that the B-ring of pregnenolone (**6**) was first hydroxylated at C-7 (probably both α and β) and then underwent oxidation. The double bond of 11α-hydroxy-7-oxopregnenolone (**13**) was then epoxidated to give 5α,6α-epoxy-3β,11α-dihydroxypregnan-7,20-dione (**14**).

After a 14-day biotransformation of pregnenolone in the culture of *Isaria farinosa* KCh KW1.1, 12.3 mg of 5α,6α-epoxy-3β,11α-dihydroxypregnan-7,20-dione (**14**) and 4.7 mg of 11α-hydroxy-7-oxopregnenolone (**13**) were isolated.

In the ^1^H NMR spectrum of compound **13**, in addition to the signal at 3.67 ppm from the proton at hydroxyl group 3-H, there is also a signal at 4.15 ppm (td, *J* = 10.6, 4.9 Hz), indicating the presence of another hydroxyl group. The analysis of the position and multiplicity of this signal and the couplings seen on the HMQC spectrum suggest that the hydroxyl group was introduced into the 11α position. At the same time, the ^13^C NMR spectrum shows the signal at 209.38 ppm, in the position characteristic of signals from carbon atoms of the carbonyl group (Table 1). The HMBC spectrum shows the coupling of the signal from this carbon with signals from protons 8-H and 9-H. At the same time, the position of the signal from the 6-H proton located at the double bond of this compound is visible at 5.76 ppm (doublet) in the position characteristic for α,β-unsaturated ketones. This confirms the location of this carbonyl group at the 7-C carbon. Based on these data, this compound can be unambiguously identified as 11α-hydroxy-7-oxopregnenolone (**13**). The main product formed during the transformation of pregnenolone (**6**) in the culture of the examined strain was compound **14**. In the ^1^H NMR spectrum performed for this compound, signals characteristic for protons bonded to the carbon at which the hydroxyl group is located at the 3β and the 11α positions were observed. At the same time, on the spectrum, the signal from the 6-H proton is visible at 3.19 ppm (singlet) in the position characteristic for epoxides. The HMBC spectrum shows the coupling of the signal from this proton with the signals from carbons 5-C and 7-C and weaker signals from carbons 4-C and 8-C. In the ^13^C spectrum, the signal from carbon 5-C is located at 72.98 ppm, which confirms the presence of the epoxy group in the structure of this product. On the other hand, the signal from carbon 7-C is located at 209.09 ppm, indicating the carbonyl bond at this carbon. Taking into account the structure of compound **13,** it can be concluded that the main product obtained during biotransformation of pregnenolone (**6**) in the culture of the examined *Isaria farinosa* KCh KW1.1 strain is 5α,6α-epoxy-3β,11α-dihydroxypregnan-7,20-dione (**14**).

In addition, to highlight the applicability of biotransformation in obtaining compounds with potential medical effects, we conducted a computational analysis of the obtained compounds and compared them with their parent compound. For this purpose, we used the free online tool SwissADME (Available online: http://www.swissadme.ch (accessed on 1 June 2022)), developed and maintained by the Molecular Modeling Group of the Swiss Institute of Bioinformatics (SIB) [22].

Table 2 shows selected information regarding physicochemical properties, lipophilicity, water solubility, pharmacokinetics and drug likeness. Complete data generated by this tool are in the Appendix A.

## 3. Discussion

Over 70 years of studies show that progesterone’s biotransformation by filamentous fungi may go in two directions: hydroxylation or Baeyer–Villiger oxidation or, rarely, a combination of them. Hydroxylation occurs mainly at the 6β and/or 11α positions [1,4,5,24,29,30,31,32,33,34,35,36,37], but hydroxylation at the 14α [29,31,33,34,35,36,38,39], 7α [33,34,35,38,39], 7β [34], 11β [30,38], 17α [24,30,32], 9α [31], 8β [31], 16α [32], 15α [32,33,40], 15β [33,34], 21 [1,36] and 12β [24,25] positions is also possible in the fungal cultures. The Baeyer–Villiger oxidation of progesterone is possible for biotransformation by some *Aspergillus* or *Penicillium* species [41,42,43,44].

The strain *Isaria farinosa* KCh KW1.1 has no ability to carry out the Baeyer–Villiger oxidation of steroid compounds, which is also confirmed by androstanes’ transformation [9]. The product with the lactone group was isolated only during the transformation of acetyloxy-5α chloro-6,19-oxidoandrostane-17-one. However, in this case, the substrate underwent sequential hydroxylation at the 19-C carbon and then further oxidation, leading to the formation of the lactone ring but without the participation of the Baeyer–Villiger-type enzymes [16]. However, this strain is capable of hydroxylating selected positions: 6β, 11α, 12β in 3-oxo-4-ene-steroids and 7α, 7β and 11α in 3β-hydroxy-5-ene derivatives. The hydroxylation position in the B ring is closely related to the arrangement of the A and B rings. The tested 3-oxo-4-ene pregnanes (**1**, **2–5**) were hydroxylated at the 6β position, while 3-hydroxy-5-ene pregnane (pregnenolone, (**6**)) was hydroxylated at the C-7 position. This correlation was also observed for the 3-oxo-4-ene (adrenosterone) and 3-hydroxy-5-ene (dehydroepiandrosterone) androstanes [9] and results from the activating vicinity of π electrons.

Hydroxylation of the C-ring of progesterone and its derivatives occurs mainly at the 11α position [1,4,5,24,29,30,31,32,33,34,35,36,37], and only a few strains can introduce a hydroxyl group into the 12β place. In the culture of *Fusarium culmorum*, progesterone was hydroxylated at the 15α position and then at 12β, giving 12β,15α-dihydroxyprogesterone [24,25]. Additionally, 12β-hydroxylation was observed for *Cephalosporium aphidicola* biotransformation but only when progesterone was hydroxylated at the 17α position or 17α-hydroxyprogesterone was a substrate [24]. In that study, no other tested progesterone derivatives were hydroxylated at the 12β position. In the above-mentioned transformations, neither 6β hydroxylation nor further hydroxylation of 12β,17α-dihydroxyprogesterone occurred. In contrast, in our study, we observed products of monohydroxylation (6β,17α-dihydroxyprogesterone) and dihydroxylation (6β,12β,17α-trihydroxyprogesterone) of 17α-hydroxyprogesterone. Similarly to the transformation catalysed by *Cephalosporium aphidicola,* we observed 12β-hydroxylation only for 17α-hydroxyprogesterone.

6β,11α-Dihydroxyprogesterone is a common product of progesterone dihydroxylation and may be obtained, among other products, e.g., in the cultures of *Aspergillus nidulans* VKPM F-1069 [5], *A. niger* N402 [1], *A. ochraceus* [37], *Rhizomucor pusillus* and *Absidia griseolla* var. *igachii* [35], *Rhizomucor tauricus* IMI23312 [45], *Cephalosporium aphidicola* [24] and *Beauveria bassiana* KCH 1065 [46]. Progesterone transformation, where 6β-hydroxy-11-oxoprogesterone (**7**) was isolated, was reported only for *Mucor* 881 (M881) [47].

16α,17α-epoxyprogesterone is a crucial intermediate in the synthesis of corticosteroids from diosgenin. Its 11α hydroxylation is a key step in synthesising prednisone and prednisolone or dexamethasone, and *Rhizopus nigricans* is the most widely used microorganism for industry purposes [48]. Other possible hydroxylation positions of this compound are 8β [49], 11β [50], 15β [49] and 15α [51]. A similar transformation to 6β,11α-dihydroxy-16α,17α-epoxyprogesterone was described for *Aspergillus ochraceus* ATCC 1008 [52].

Interestingly, the limited positions of androstane hydroxylation are reflected in the low number of pregnane hydroxyl derivatives. In many cases, the number of progesterone hydroxylation products significantly exceeds the number of hydroxylation products of androstanes obtained by biotransformation by the same strain, e.g., two products from androstenedione and eight from progesterone transformation by *Didymosphearia igniaria* KCH 6670 [34]. Further, many progesterone biotransformations result in many products, like in the culture of *Cephalosporium aphidicola* [24]. In the culture of the *Isaria fumosorosea* KCh J2 strain, progesterone was transformed into so many products that their purification and identification were impossible [10]. In this study, we observed only mono- or dihydroxylation products, and the number of obtained products did not exceed three.

Pregnenolone results from cholesterol side-chain cleavage and the precursor of all steroid hormones in mammals. In the human body, the isaria fumoso 3-hydroxy-5-ene moiety of pregnenolone (and 17β-hydroxypregnenolone) is transformed to 3-oxo-4-ene of progesterones. The same oxidation and isomerisation pathway can be observed in fungal transformation [34,53,54]. Kollerov and others tested 75 selected fungal strains of *Ascomycota* and *Zygomycota* divisions to transform pregnenolone, especially into 11α-hydroxyprogesterone [54]. In total, 60 strains converted pregnenolone, but only 21 fungal strains were able to form 7α-hydroxypregnenolone, thus evidencing the presence of 7α-hydroxylase activity. Ten tested fungal strains transformed pregnenolone to 7β,11α-dihydroxypregnenolone, but in these cases, the authors did not detect any monohydroxylated products, neither 7β- nor 11α-hydroxypregnenolone. Other products were not determined because of their low yield. Additionally, 20 tested strains were characterised as transforming pregnenolone to progesterone, indicating the presence of dual-function 3-hydroxysteroid dehydrogenase (3β-hydroxyl function oxidation/Δ^5−4^ isomerisation). In our study, we did not detect mono-hydroxylation derivatives of pregnenolone or even 7,11α-dihydroxypregnenolone. We aimed to provide evidence of the transformation of steroids by obtaining products in a sufficient amount for spectroscopic analysis (NMR), which is why we conducted the experiment in the control of TLC and completed the reaction by extraction after total transformation of the substrate. By that time, the intermediates may have undergone a further transformation and thus remained in the crude medium in quantities that prevented their identification. We did not provide transformation aiming at the isolation of these possible intermediates; thus, we did not propose here the potential pathway for this biotransformation, including intermediates, only showing isolated products. Like most of the tested strains in the Kollerov study, *Isaria farinosa* KCh KW1.1 cannot isomerise pregnenolone to progesterone. The lack of 3-hydroxysteroid dehydrogenase was also confirmed in the transformation of 3β-hydroxy-5-ene androstane—dehydroepiandrosterone [9]. Compound **13** is mentioned in a patent application as a product of pregnenolone transformation in the culture of *Rhizopus arrhizus* [55]. In the current literature, **14** was not reported.

To highlight the applicability of biotransformation and the high value of the obtained compounds, we used a SwissADME tool to provide computational analysis for rapid prediction of key parameters for drug discovery. The rule of five, formulated by Christopher Lipinski in 1997, gives some clues as to what a compound should look like to be orally administered. These criteria are:-no more than five hydrogen bond donors (expressed as the sum of OHs and NHs);-no more than 10 hydrogen bond acceptors (expressed as the sum of Ns and OS);-a molecular mass less than 500 Da;-an octanol–water partition coefficient (log P) that does not exceed 5 [56].

Considering the relevant data in this field, summarised in Table 2, we can see that none of the obtained compounds exceeds the molecular weight of 500 Da (or g/mol). The molecular weight of progesterone (**1**), the basic compound in this set, is 314.46 Da. Modifications of the tested compounds consist of leading, at most, three additional oxygen atoms, which does not significantly increase the mass of the initial compound. The mass of the heaviest compounds in this set is 362.46 Da (compounds **10** and **14**).

Significantly increasing the molecule mass impacts its worse intestinal and blood–brain barrier permeability. Of course, this is not the only determinant of membrane permeation, and the others are lipophilicity and polarity. Lipophilicity is calculated as the partition coefficient between n-octanol and water (log Po/w), and polarity is considered as a surface sum over all polar atoms of the molecule (oxygen, nitrogen, sulfur and phosphorus), including their attached hydrogen atoms [22]. In our case, the impact on those two factors has additional oxygen atoms. In Figure 6, we summarised the results for lipophilicity (XLOGP3), polarity (TPSA) and insolubility (LOG S (ESOL)) of the tested compounds. According to the SwissADME, the most promising values for lipophilicity, polarity and insolubility are in the range: −0.7 < XLOGP3 < 5.0; 20A^2^ < TPSA < 130 A^2^; −6 < LOG S (ESOL) < 0, respectively. All the tested compounds are within these values; however, it is easy to see the correlations between these factors and the structure of the compound. Progesterone is the less polar compound (compound **1**), and with the addition of hydrogen atoms (as hydroxyl or carbonyl groups), the polarity of the compounds increases. In each biotransformation, the obtained compounds are more polar than the substrate (Figure 6).

In simple terms, polarity is related to solubility, following the principle that “like dissolves like”. Polar solvents dissolve polar solutes and, conversely, non-polar solvents dissolve non-polar solutes. However, this simple rule is more complicated. The authors of the estimated solubility (ESOL) method tested nine parameters, but only four made a significant contribution to the model [28]. The authors took into consideration only the log Po/w, molecular weight (MW), number of rotatable bonds (RB) and aromatic proportion (AP) and gave log Po/w and AP the most value. The final result, represented as log S, is the solubility unit corresponding to the 10-based logarithm of the solubility of a molecule measured in mol/L, so a less negative number means better uptake or distribution of biologically active compounds. The value of log S for progesterone (**1**) is −4.16, but the transformation changes this factor to −2.56 and −2.62 for compounds **2** and **7**, respectively. Similar changes were observed for compounds **5** and **6** (Appendix A).

All values for lipophilicity, polarity and insolubility are gathered in Table 2, but for easier tracking of these correlations, we provide Figure 6.

The SwissADME tool provides one more calculation called BOILED-Egg (Figure 7). This is a graphical result of simultaneous prediction of the passive gastrointestinal absorption (HIA) and brain access (BBB). The innermost part, the yolk, shows compounds with highly probable BBB permeation, and the middle part, the white, shows molecules with highly likely HIA absorption. Outside the egg is the grey part, which is responsible for compounds with predicted low absorption and limited brain penetration. The tool built this graph based on two parameters, log P and TPSA, for lipophilicity and apparent polarity, and it only models passive penetration through a gastro-intestinal wall and blood–brain barrier. Additionally, the model predicts that the tested compounds may be efficiently removed from these compartments by the P-glycoprotein. Analysing that graph, we can observe that most of the obtained compound is predicted to be removed by P-glycoprotein from the gastrointestinal tract and the central nervous system (Appendix A).

## 4. Materials and Methods

### 4.1. Materials

The substrates, progesterone (**1**), 11α-hydroxyprogesterone (**3**), 17α-hydroxyprogesterone (**4**), 16α,17α-epoxyprogesterone (**5**) and pregnenolone (**6**) were purchased from Sigma-Aldrich.

The biocatalyst was *Isaria farinosa* KCh KW1.1, a strain obtained from the Department of Chemistry collection, Wrocław University of Environmental and Life Sciences, Poland. Its isolation and genetic determination were described previously [9].

### 4.2. Biotransformation

An amount of 500 mL of the sterile cultivation medium (3% glucose and 1% aminobac dissolved in distilled water) in Erlenmeyer flasks (2000 mL) was inoculated with a suspension of *Isaria farinosa* KCh KW1.1 cells and then incubated for three days at 25 °C on a rotary shaker. Then, 100 mg of an appropriate substrate (**1**, **3**–**6**) dissolved in 1 mL of DMSO was added. The transformation was carried out under the control of TLC. When the substrate was not visible under the UV light after spraying it with H_2_SO_4_/MeOH solution, the medium was extracted with chloroform (3 × 300 mL), dried (MgSO_4_) and concentrated in vacuo. The transformation products were separated with preparative TLC and analysed using spectroscopic methods (NMR) (Appendix A).

### 4.3. Analytical Methods

Products were separated using the preparative TLC plates (Silica Gel GF, 20 × 20 cm, 500 μm, Analtech) and a hexane/acetone mixture (2:1, *v*/*v*) as an eluent. The NMR spectra were recorded on a DRX 500 MHz Bruker spectrometer and measured in CDCl_3_ or DSMO-*d*_6_ for low solubility in chloroform products (Appendix A).

**Β,11α-dihydroxyprogesterone (2)** ^1^H NMR (600 MHz) (ppm) (CDCl_3_) δ: 0.73 (s, 3H, 18-H); 1.10 (t, 1H, *J* = 10.3 Hz, 9-H); 1.27–1.36 (m, 3H, 7-Hα, 14-H, 15-Hβ); 1.50 (s, 3H, 19-H); 1.51 (t, 1H, *J* = 11.4 Hz, 12-Hα); 1.67–1.78 (m, 2H, 15-Hα, 16-Hα); 1.92 (td, 1H, *J* = 14.4, 4.4 Hz, 1-Hα); 1.99 (dt, 1H, *J* = 13.9, 3.0 Hz, 7-Hβ,); 2.01–2.07 (qd, 1H, *J* = 11.9, 2.7 Hz, 8-H); 2.13 (s, 3H, 21-H); 2.16–2.23 (m, 1H, 16-Hβ); 2.28–2.40 (m, 2H, 2-Hα, 12-Hβ); 2.51–2.58 (m, 2H, 2-Hβ, 17-Hα); 2.79 (ddd, 1H, *J* = 13.7, 4.8, 3.0 Hz, 1-Hβ); 4.09 (td, 1H, *J* = 10.6, 4.8 Hz, 11-Hβ); 4.34 (t, 1H, *J* = 2.8 Hz, 6-Hα); 5.81 (s, 1H, 4-H). ^1^H NMR (600 MHz) (ppm) (DSMO-*d*_6_) δ: 0.58 (s, 3H, 18-H); 0.98 (t, 1H, *J* = 10.4 Hz, 9-H); 1.12 (ddd, 1H, *J* = 17.9, 11.9, 5.9 Hz, 15-Hβ); 1.17 (td, 1H, *J* = 14.1, 2.2 Hz, 7-Hα); 1.23 (td, 1H, *J* = 11.7, 6.3 Hz, 14-H); 1.37 (s, 3H, 19-H); 1.46 (t, 1H, *J* = 11.5 Hz, 12-Hα); 1.55–1.64 (m, 2H, 15-Hα, 16-Hα); 1.73–1.81 (m, 2H, 1-Hα, 7-Hβ); 1.88 (qd, 1H, *J* = 11.2, 2.5 Hz, 8-H); 2.00–2.07 (m, 1H, 16-Hβ); 2.07 (s, 3H, 21-H); 2.13 (dt, 1H, *J* = 16.9, 3.3 Hz, 2-Hα); 2.18 (dd, 1H, *J* = 12.0, 4.6 Hz, 12-Hβ); 2.43 (ddd, 1H, *J* = 17.1, 14.9, 4.7 Hz, 2-Hβ); 2.60 (t, 1H, *J* = 8.8 Hz, 17-Hα); 2.74 (dt, 1H, *J* = 13.8, 3.9 Hz, 1-Hβ); 3.85 (tdd, 1H, *J* = 10.8, 6.0, 4.8 Hz, 11-Hβ); 4.13 (q, 1H, *J* = 2.8 Hz, 6-Hα); 4.41 (dd, 1H, *J* = 6.4, 2.5 Hz, 11-O*H*); 5.12 (s, 1H, 6-O*H*); 5.64 (s, 1H, 4-H).

**6β,17α-dihydroxyprogesterone (8)** was isolated in a small amount. Spectral data are identical to those described in the literature [23,24,25].

**12β,17α-dihydroxyprogesterone (9)** ^1^H NMR (600 MHz) (ppm) (CDCl_3_) δ: 0.84 (s, 3H, 18-H); 1.02–1.10 (m, 2H, 7-Hα, 9-H); 1.18 (s, 3H, 19-H); 1.39 (td, 1H *J* = 12.9, 11.2 Hz, 11-Hβ); 1.48 (td, 1H, *J* = 12.0, 6.0 Hz, 15-Hβ); 1.54 (ddd, 1H, *J* = 22.4, 11.3, 3.5 Hz, 16-Hα); 1.67 (td, 1H, *J* = 11.5, 7.7 Hz, 14-H); 1.70 (td, 1H, *J* = 13.9, 4.8 Hz, 1-Hα); 1.75–1.83 (m, 2H, 11-Hα, 16-Hβ); 1.83–1.92 (m, 2H, 7-Hβ, 15-Hα); 2.02 (ddd, 1H, *J* = 13.3, 4.9, 3.0 Hz, 1-Hβ); 2.30 (ddd, 1H, *J* = 14.6, 4.1, 2.4 Hz, 2-Hα); 2.39 (s, 3H, 21-H); 2.34–2.42 (m, 3H, 2-Hβ, 6-Hα, 8-H); 2.46 (ddd, 1H, *J* = 14.6, 11.5, 2.8 Hz,6-Hβ); 3.94 (dd, 1H, *J* = 11.1, 4.8 Hz, 12-Hα); 5.73 (s, 1H, 4-H).

**6β,12β,17α-trihydroxyprogesterone (10)** ^1^H NMR (600 MHz) (ppm) (CDCl_3_) δ: 0.89 (s, 3H, 18-H); 1.05 (ddd, 1H, *J* = 12.9, 10.9, 3.9 Hz, 9-H); 1.45 (q, 1H, *J* = 12.8 Hz, 7-Hα); 1.51 (s, 3H, 19-H); 1.54 (qd, 1H, *J* = 11.9, 5.9 Hz, 15-Hβ); 1.68 (ddd, 1H, *J* = 11.9, 8.0, 4.2 Hz, 14-H); 1.74 (td, 1H, *J* = 13.9, 4.8 Hz, 1-Hα); 1.78–1.86 (m, 2H, 11-Hα, 16-Hα); 1.88–1.95 (m, 1H, 15-Hα); 1.96–2.08 (m, 3H, 1-Hβ, 7-Hβ, 16-Hβ); 2.40 (s, 3H, 21-H); 2.41 (ddd, 1H, *J* = 14.5, 4.0, 2.4 Hz, 2-Hα); 2.47–2.55 (m, 2H, 2-Hβ, 8-H); 3.98 (dd, 1H, *J* = 11.1, 4.8 Hz, 12-Hα); 4.37 (t, 1H, *J* = 2.6 Hz, 6-Hα); 5.83 (s, 1H, 4-H).

**6β-hydroxy-16α,17α-epoxyprogesterone (11)** was isolated in a small amount. The spectral data are identical to those reported previously in the literature [26].

**6β,11α-dihydroxy-16α,17α-epoxyprogesterone (12)** ^1^H NMR (600 MHz) (ppm) (CDCl_3_) δ: 1.04 (t, 1H, *J* = 10.1 Hz, 9-H); 1.09 (s, 3H, 18-H); 1.23–1.35 (m, 3H, 7-Hα, 14-H, 15-Hβ); 1.42 (t, 1H, *J* = 11.6 Hz, 12-Hα); 1.48 (s, 3H, 19-H); 1.94–1.85 (m, 2H, 1-Hα, 7-Hβ); 1.97 (dd, 1H, *J* = 13.1, 6.0 Hz, 15-Hα); 2.01 (s, 3H, 21-H); 2.12 (qd, 1H, *J* = 11.9, 2.7 Hz, 8-H); 2.32 (dt, 1H, *J* = 17.4, 3.3 Hz, 2-Hβ); 2.39 (dd, 1H, *J* = 11.8, 5.1 Hz, 12-Hβ); 2.50 (ddd, 1H, *J* = 17.5, 14.7, 3.2 Hz, 2-Hα); 2.71 (ddd, 1H, *J* = 13.8, 4.4, 3.2 Hz, 1-Hβ); 3.72 (s, 1H, 16-Hβ); 4.10 (td, 1H, *J* = 10.2, 5.2 Hz, 11-Hβ); 4.31 (t, 1H, *J* = 2.7 Hz, 6-Hα); 5.77 (s, 1H, 4-H). ^1^H NMR (600 MHz) (ppm) (DSMO-*d*_6_) δ: 0.92 (t, 1H, *J* = 10.2 Hz, 9-H); 1.00 (s, 3H, 18-H); 1.10 (td, 1H, *J* = 12.1, 6.4 Hz, 14-H); 1.13–1.19 (m, 1H, 7-Hα); 1.22–1.33 (m, 2H, 12-Hα, 15-Hβ); 1.38 (s, 3H, 19-H); 1.71 (dt, 1H, *J* = 13.3, 2.8 Hz, 7-Hβ); 1.78 (td, 1H, *J* = 14.4, 4.1 Hz, 1-Hα); 1.86 (dd, 1H, *J* = 13.4, 6.3 Hz, 15-Hα); 1,97 (s, 3H, 21-H); 2.01 (qd, 1H, *J* = 11.9, 2.6 Hz, 8-H); 2.13 (dt, 1H, *J* = 17.4, 3.4 Hz, 2-Hβ); 2.19 (dd, 1H, *J* = 11.9, 5.1 Hz, 12-Hβ); 2.42 (ddd, 1H, *J* = 17.1, 14.8, 4.7 Hz, 2-Hα); 2.71 (ddd, 1H, *J* = 13.8, 4.0, 3.7 Hz, 1-Hβ); 3.86–3.91 (m, 1H, 11-Hβ); 3.95 (s, 1H, 16-Hβ); 4.31 (q, 1H, *J* = 2.7 Hz, 6-Hα); 4.37 (d, 1H, *J* = 6.6 Hz, 11-O*H*); 5.13 (d, 1H, *J* = 2.3 Hz, 6-O*H*); 5.64 (s, 1H, 4-H).

**11α-hydroxy-7-oxopregneonolone (13)**^1^H NMR (600 MHz) (ppm) (CDCl_3_) δ: 0.68 (s, 3H, 18-H); 1.23–1.33 (m, 2H, 1-Hα, 16-Hα); 1.35 (s, 3H, 19-H); 1.39 (t, 1H, *J* = 11.4 Hz, 9-H); 1.51–1.59 (m, 2H, 12-Hα, 14-Hα); 1.63–1.69 (m, 1H, 2-Hα); 1.83–1.93 (m, 2H, 2-Hβ, 15-Hβ); 2.10–2.13 (m, 1H, 15-Hα); 2.14 (s, 3H, 21-H); 2.29 (t, 1H, *J* = 6.5 Hz, 8-Hα); 2.35 (dd, 1H, *J* = 11.7, 5.1 Hz, 12-Hβ); 2.42 (ddd, 1H, *J* = 13.2, 11.3, 1.6 Hz, 4-Hα); 2.48 (t, 1H, *J* = 9.5 Hz, 17-Hα); 2.50–2.56 (m, 2H, 4-Hβ, 16-Hβ); 2.71 (dt, 1H, *J* = 14.2, 3.5 Hz, 1-Hβ); 3.68 (tt, 1H, *J* = 11.3, 4.8 Hz, 3-Hα); 4.15 (td, 1H, *J* = 10.6, 4.9 Hz, 11-Hβ); 5.76 (d, 1H, *J* = 1.4 Hz, 4-H).

**5α,6α-epoxy-3β,11α-dihydroxypregnan-7,20-dione****(14)**^1^H NMR (600 MHz) (ppm) (CDCl_3_) δ: 0.66 (s, 3H, 18-H); 1.00–1.09 (m, 1H, 15-Hβ); 1.18–1.34 (m, 3H, 1-Hα, 12-Hα, 14-H); 1.38 (s, 3H, 19-H); 1.39 (ddd, 1H, *J* = 13.5, 4.8, 2.5 Hz, 4-Hβ); 1.43–1.57 (m, 3H, 2-Hα, 9-H, 16-Hα); 1.76–1.88 (m, 2H, 2-Hβ, 15-Hα); 2.11 (s, 3H, 21-H); 2.08–2.11 (m, 2H, 4-Hα, 16-Hβ); 2.26 (dd, 1H, *J* = 11.7, 4.7 Hz, 12-Hβ); 2.48 (t, 1H, *J* = 9.3 Hz, 17-Hα); 2.58 (dt, 1H, *J* = 14.4, 3.5 Hz, 1-Hβ); 2.99 (dd, 1H, *J* = 12.9, 11.2 Hz, 8-H); 3.19 (s, 1H, 6-Hβ); 3.75 (tt, 1H, *J* = 11.2, 4.6 Hz, 3-Hα); 4.07 (td, 1H, *J* = 10.5, 4.6 Hz, 11-Hβ).

### 4.4. Computational Analysis

Analysis was performed using the free online tool SwissADME (Available online: http://www.swissadme.ch (accessed on 1 June 2022)), developed and maintained by the Molecular Modeling Group of the Swiss Institute of Bioinformatics (SIB) [22]. All data were collected on 1 June 2022 (Appendix A).

Compounds **1**–**14** were drawn one by one in the molecular sketcher and then transformed to SMILES. The calculations were run when all **14** SMILES were in the SMILES list. Then, the results were imported as a CSV file from the website. Additionally, graphical results for all calculations were copied.

## 5. Conclusions

*Isaria farinosa* is a well-known fungus, primarily because of its entomopathogenic abilities. So far, no data on the ability of *Isaria farinosa* strains to convert steroids have been published, so we examined this ability. In our previous study, we screened 12 strains of this genus for the transformation of selected steroid compounds, e.g., dehydroepiandrosterone, androstenediol (androst-5-ene-3β,17β-diol), androstenedione (androst-4-ene-3,17-dione), adrenosterone (androst-4-ene-3,11,17-trione, progesterone and 17α-methyltestosterone) [8]. In this study, we focused on progesterone and its derivatives.

These study results indicate high stereo- and regioselectivity of the *Isaria farinosa* KCh KW1.1 strain. All tested 3-oxo-4-ene derivatives (progesterone (**1**), 11α-hydroxyprogesterone (**3**), 17α-hydroxyprogesterone (**4**), 16α,17α-epoxyprogesterone (**5**)) were hydroxylated at the 6β position. On the other hand, pregnenolone (3β-hydroxy-5-ene arrangement) was hydroxylated at C-7. This difference was caused by activating the vicinity of π electrons. Excluding 17α-hydroxyprogesterone (**4**), all tested compounds were also hydroxylated at the 11α position, important in steroid drug synthesis. 17α-Hydroxyprogesterone (**4**) was hydroxylated at the 12β position.

The number of the obtained derivatives is worth emphasising. In this study, we observed only the mono- and dihydroxy derivatives of the tested substrates, and the number of the obtained products for each biotransformation did not exceed three.

Biotransformation of progesterone derivatives confirmed the lack of 3-hydroxysteroid dehydrogenase and an inability for a Baeyer–Villiger oxidation by the *Isaria farinose* KCh KW1.1 strain, which was noticeable in androstane transformations [8].

The results demonstrate a high biocatalytic potential of *Isaria farinosa* KCh KW1.1 strain towards progesterone and pregnenolone, revealing the existence of a novel active biocatalyst that can be exploited for the synthesis of high-value hydroxysteroids and expanding the knowledge on the biocatalytic potential of entomopathogenic fungi.

Computational analyses were calculated with the free online tool SwissADME (Available online: http://www.swissadme.ch (accessed on 1 June 2022)), developed and maintained by the Molecular Modeling Group of the Swiss Institute of Bioinformatics (SIB). The analysis of the obtained compounds revealed that biotransformation methods may be helpful in improving pharmacokinetic parameters, such as the polarity and solubility of active compounds.

## Figures and Tables

**Figure 1 ijms-23-07015-f001:**
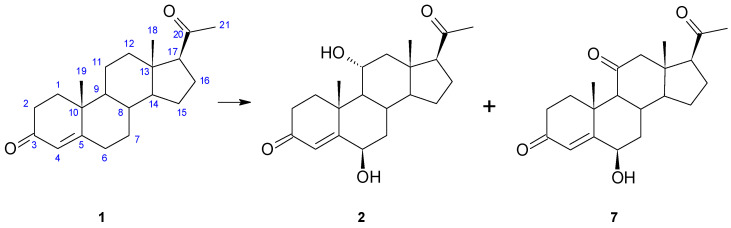
Transformation of progesterone (**1**) in the culture of *Isaria farinosa* KCh KW1.1.

**Figure 2 ijms-23-07015-f002:**
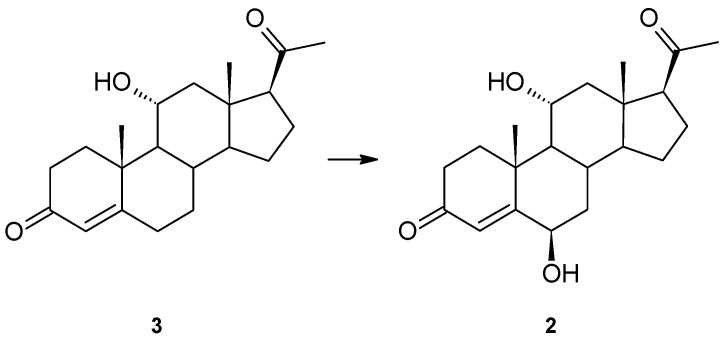
Hydroxylation of 11α-hydroxyprogesterone (**3**) by *Isaria farinosa* KCh KW1.1.

**Figure 3 ijms-23-07015-f003:**
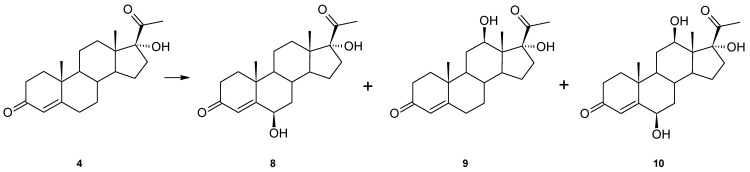
Biotransformation of 17α-hydroxyprogesterone (**4**) in *Isaria frinose* KCh KW1.1 culture.

**Figure 4 ijms-23-07015-f004:**
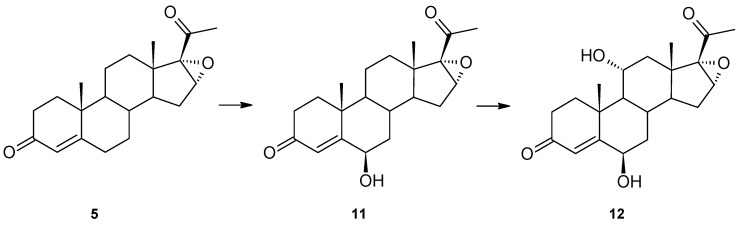
Transformation of 16α,17α-epoxyprogesterone (**5**) by *Isaria farinosa* KCh KW1.1.

**Figure 5 ijms-23-07015-f005:**
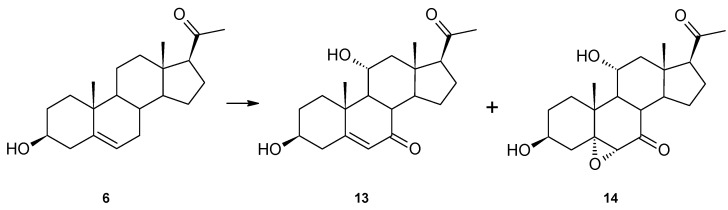
Pregnenolone (**6**) modification by *Isaria farinosa* KCh KW1.1 strain.

**Figure 6 ijms-23-07015-f006:**
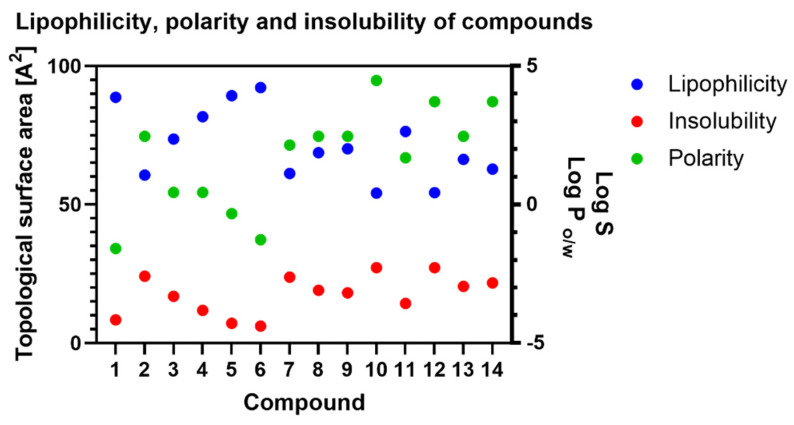
Changes in lipophilicity, polarity and insolubility of tested compounds. Lipophilicity is measured as a Log P parameter, polarity as a topological surface area (TPSA) and insolubility as a Log S value.

**Figure 7 ijms-23-07015-f007:**
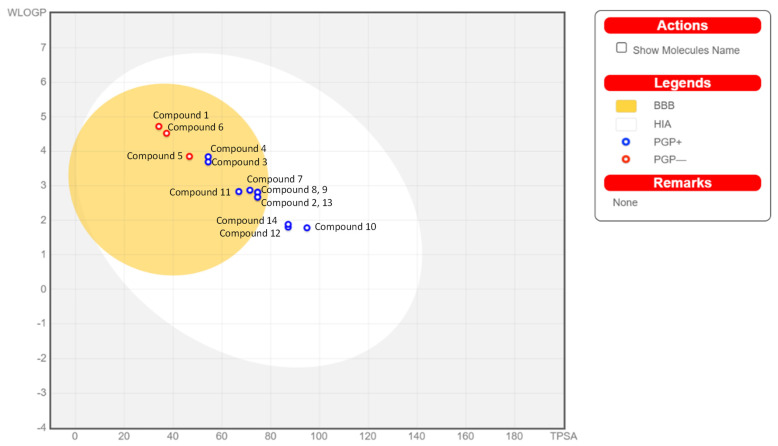
BOILED-Egg analysis (described in the text).

**Table 1 ijms-23-07015-t001:** Measured positions of signals visible in the ^13^C NMR spectrum.

Atom Number	Products							
	2 ^a^	2 ^b^	9 ^a^	10 ^a^	12 ^a^	12 ^b^	13 ^a^	14 ^a^
1	39.11	38.65	35.79	37.16	38.82	38.41	38.48	38.11
2	34.55	34.14	32.45	34.31	34.53	34.10	31.47	30.84
3	201.14	199.88	199.56	200.23	201.04	199.77	70.87	68.89
4	127.17	125.61	124.33	126.83	127.27	125.71	42.50	41.45
5	168.41	169.74	170.39	167.48	168.26	169.43	165.23	72.98
6	73.19	71.29	34.54	73.14	73.04	71.22	125.48	64.80
7	37.66	38.03	31.45	37.99	37.41	37.78	200.84	209.09
8	28.50	28.03	34.00	34.41	26.38	25.86	44.67	42.07
9	59.10	58.26	52.02	51.91	59.52	58.89	49.12	60.80
10	39.39	38.96	38.55	37.99	39.54	39.12	40.39	36.4
11	69.02	67.23	29.37	29.84	68.55	66.79	68.51	68.89
12	50.52	49.39	70.08	70.16	43.25	42.68	49.09	48.84
13	44.40	43.72	53.05	53.43	41.92	40.43	44.42	43.85
14	55.44	54.79	48.54	48.59	44.33	43.99	55.56	48.10
15	24.32	23.93	23.80	23.85	27.33	26.70	24.01	24.08
16	23.14	22.31	34.44	28.74	60.78	60.46	26.10	23.74
17	63.27	62.45	89.20	89.22	70.37	69.94	61.99	62.29
18	14.67	14.27	10.15	10.30	16.48	16.00	14.39	14.48
19	20.33	19.67	17.38	19.62	20.29	19.66	17.19	15.91
20	209.06	208.42	214.40	214.76	204.63	204.72	209.38	209.13
21	31.45	31.07	28.16	28.31	25.89	25.75	31.57	31.43

^a^ dissolved in CDCl_3_; ^b^ in DMSO-*_d_*_6._

**Table 2 ijms-23-07015-t002:** Selected parameters obtained by the SwissADME web tool.

Compound	MW ^a^	Fraction Csp3 ^b^	#Rotatable Bonds ^c^	TPSA ^d^	XLOGP3 ^e^	ESOL Log S ^f^	Lipinski #Violations ^g^	GI Absorption ^h^	BBB Permeant ^i^	Pgp Substrate ^j^
**1**	314.46	0.81	1	34.14	3.87	−4.16	0	High	Yes	No
**2**	346.46	0.81	1	74.60	1.06	−2.59	0	High	Yes	Yes
**3**	330.46	0.81	1	54.37	2.36	−3.31	0	High	Yes	Yes
**4**	330.46	0.81	1	54.37	3.17	−3.82	0	High	Yes	Yes
**5**	328.45	0.81	1	46.67	3.93	−4.29	0	High	Yes	No
**6**	316.48	0.86	1	37.30	4.22	−4.39	0	High	Yes	No
**7**	344.44	0.76	1	71.44	1.12	−2.62	0	High	Yes	Yes
**8**	346.46	0.81	1	74.60	1.87	−3.10	0	High	Yes	Yes
**9**	346.46	0.81	1	74.60	2.01	−3.19	0	High	Yes	Yes
**10**	362.46	0.81	1	94.83	0.41	−2.28	0	High	No	Yes
**11**	344.44	0.81	1	66.90	2.64	−3.57	0	High	Yes	Yes
**12**	360.44	0.81	1	87.13	0.43	−2.28	0	High	No	Yes
**13**	346.46	0.81	1	74.60	1.63	−2.95	0	High	Yes	Yes
**14**	362.46	0.90	1	87.13	1.28	−2.83	0	High	No	Yes

Notes: ^a^ Molecular weight; ^b^ saturation of compound described as the ratio of sp3 hybridised carbons over the total carbon count of the molecule; ^c^ the number of bonds that allow free rotation around themselves, which are defined as any single bond, not in a ring, bound to a nonterminal heavy atom; ^d^ the topological polar surface area—the surface sum over all polar atoms of the molecule (oxygen, nitrogen, sulfur and phosphorus), including their attached hydrogen atoms; ^e^ the logarithm of the partition coefficient between n-octanol and water (logP), XLOGP3 predicts the log P value of a query compound by using the known log P value of a reference compound as a starting point [27]; ^f^ the aqueous solubility (ESOL—estimated solubility) of a compound estimated directly from its structure, based on calculated log P_octanol_, molecular weight, the proportion of heavy atoms in aromatic systems and number of rotatable bonds [28]; ^g^ number of parameters above the Lipinski filter: MW ≤ 500, MLOGP ≤ 4.15, N or O ≤ 10, NH or OH ≤ 5; ^h^ gastrointestinal absorption according to the white of BOILED-Egg; ^i^ blood–brain barrier permeation according to the yolk of BOILED-Egg; ^j^ P-glycoprotein substrate.

## Data Availability

Not applicable.

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
