# Peer review of "Hydroxylation of Progesterone and Its Derivatives by the Entomopathogenic Strain *Isaria farinosa* KCh KW1.1"

_ijms, 2022, doi:10.3390/ijms23137015_

Round 1

Reviewer 1 Report

The biotransformation of progesterone with microbes is an interesting topic. In this work, the authors investigated the fungus strain Isaria farinosa KCh KW1.1 to transform progesterone derivatives. The products were separated and analyzed with TLC and NMR. The work will be significant if the following points are provided:

  1. The TLC images of products and substrate should be presented to prove the conversion.
  2. If possible, provide the analysis of the genome sequence of the fungus strain. Explain why it has the ability to transform progesterone.

Author Response

x

Reviewer 2 Report

1. There are many enzyme systems in the fungi. Some are expressed and others are not expressed in specific culture conditions. Those reaction not seen in one culture condition may be found in other culture condition. For example, the biotransformation ability of this fungus in rich medium is different from that in minimum medium since the expressed enzyme system may not be the same. This study cannot exclude the possibility that a enzyme system capable of transformation of progesterone to other products not found in this study.

2. Whether or not the fungus lacing some hydroxylase can only be proved by genome sequencing and fully functional analysis. Since some Isaria farinose strains were sequenced, it is strong encouraged to perform investigation experiment toward finding enzymes relating to the biotransformation reaction so as to meet the interest of readers of IJMS.

3. To fit the interest of readers of IJMS, some physical-chemical properties of new compound such as solubility or stability, or even some biological activities, should be provided.

4. The biotransformation efficiency is important. It is important to provide the data of HPLC or GC analysis of each biotransformation.

5. The position shoud be numbered in the compound in the figure so as to make readers easy to realize the hydroxylation sites.

6. Some description sentences in the manuscript are ambiguous, such as line109-110, line 255-257, line 278-279.

Some biotransformation abilities toward progesterone and its derivatives by Isaria farinos KCh KW1.1 were found in this study, however, the scientific depth of current study is not enough for published in IJMS. It is recommended that submitting to MDPI journal Catalysts or Fermentation.

Author Response

x

Round 2

Reviewer 1 Report

The authors have provided all the materials and data. Now the manuscript could be accepted in the present form.

Author Response

.